



# Establishing a sediment budget in the newly created 'Kleine Noordwaard' wetland area in the Rhine-Meuse delta

Eveline Christien van der Deijl[1], Marcel van der Perk[1], and Hans Middelkoop[1]

[1]Faculty of Geosciences, Universiteit Utrecht, The Netherlands.

*Correspondence to:* E. C. van der Deijl (E.C.vanderDeijl@uu.nl)

**Abstract.** Many deltas are threatened by accelerated soil subsidence, sea-level rise, increasing river discharge, and sediment starvation. Effective delta restoration and effective river management require a thorough understanding of the mechanisms of aggradation, erosion, and their controls. Sediment dynamics has been studied at floodplains and marshes, but little is known about the sediment dynamics and budget of newly created wetlands. Here we take advantage of a recently opened tidal freshwater system to study both the mechanisms and controls of aggradation and erosion in newly created wetlands. We quantified both the magnitude and spatial patterns of aggradation and erosion in a former polder area in which water and sediment have been reintroduced since 2008. Based on terrestrial and bathymetric elevation data, supplemented with field observations of the location and height of cut banks and the thickness of the newly deposited layer of sediment, we determined the sediment budget of the study area for the period 2008-2015. Aggradation primarily took place in channels in the central part of the former polder area, whereas channels near the inlet and outlet of the area experienced considerable erosion. At the intertidal flats, sand aggradation especially takes place at low lying locations close to the channels. Mud aggradation typically occurs further away from the channels, but sediment is in general uniformly distributed over the intertidal area, due to the presence of topographic irregularities and micro topographic flow paths. Cut bank retreat does not significantly contribute to the total sediment budget, because wind wave formation is limited by the length of the fetch. Consecutive measurements of channel bathymetry show a decrease in erosion and aggradation rates over time, but the overall result of this study indicate that the area functions as a sediment trap. On average, the area traps approximately 46 % of the sediment delivered to the study area, which is approximately 3 % of the sediment load of the River Rhine at the Dutch-German border. The total sediment budget of the study area amounts to 29.7 $10^3$ $m^3$ year$^{-1}$, which corresponds to a net area-averaged aggradation rate of 5.1 mm year$^{-1}$. This is enough to compensate for the actual rates of sea-level rise and soil subsidence in The Netherlands.

## 1 Introduction

Many deltas in the world cope with drowning and loss of delta land due to flood protected polders, dams, and embankments of channels, which result in accelerated soil subsidence and sediment starvation (Ibáñez et al., 1997; Syvitski and Saito, 2007; Ibáñez et al., 2014). The urgency of this problem is enhanced by sea-level rise (Syvitski, 2008) or increasing river discharge. Most deltas are valuable and densely populated, because of their ideal location for harbours, agriculture, aquaculture, and tourism (Kirwan and Megonigal, 2013; Ibáñez et al., 2014). Moreover, they encompass vast wetland areas of great ecological





value. Traditional approaches in river management aim at reducing flood risks by constructing dikes and dams. Although such constructions are often effective in reducing flood risks, they disrupt the morphodynamic processes and ecological functioning of the system, increase sediment starvation, and involve high costs for construction and maintenance (Hudson et al., 2008). Therefore, since recently, river delta management has been shifting from the implementation of these strong regulations towards

the control of a more natural system where dynamic processes are restored and the system becomes multifunctional. Examples include the Tidal River Management project in Bangladesh (Khadim et al., 2013), the diversion projects in the Mississippi deltaic plain (DeLaune et al., 2003; Day et al., 2007; Paola et al., 2011), the Plan Integrale de Protección del Delta Ebro in the Ebro Delta (Calvo-Cubero et al., 2013) and the Room for the River initiative in the Netherlands (Rijke et al., 2012).

    Paola et al. (2011) defined river delta restoration as diverting sediment and water from major channels into adjoining drowned

areas, where the sediment can build new land and provide a platform for regenerating wetland ecosystems. Delta restoration is effective when sedimentation compensates for sea-level rise and soil subsidence. Table 1 summarizes sedimentation rates determined in various types of delta compartments representing different depositional environments. Sediment deposition is variable and complex. This table suggests that accumulation is positively related to the suspended sediment concentration. Furthermore, it is widely known from the literature that sediment deposition is controlled by the frequency and duration of

inundation (French and Spencer, 1993; Middelkoop and van der Perk, 1998; Reed et al., 1999; Temmerman et al., 2003; Thonon et al., 2007), the suspended sediment concentration in the feeding channel (Asselman and Middelkoop, 1998; French and Spencer, 1993), and the ability of sediment to settle, which is in turn controlled by vegetation (Darke and Megonigal, 2003; Temmerman et al., 2005b; Schile et al., 2014; Mitsch et al., 2014), the flow paths to or within the wetland/compartment (French and Spencer, 1993; Siobhan Fennessy et al., 1994; Reed et al., 1999; Davidson-Arnott et al., 2002; Temmerman et al.,

2003; Anderson and Mitsch, 2007; Mitsch et al., 2014), and the residence time within the compartment (Asselman and Van Wijngaarden, 2002). Although considerable research has been devoted to sediment deposition in wetlands in coastal deltas and river floodplains, remarkably few empirical field studies have been reported on the initial formation and evolution of newly created wetlands.

    In the present study, we take advantage of a recently opened tidal freshwater system to study the aggradation or erosion

in newly created tidal wetlands. In the framework of the Room for the River (RfR) initiative in the Netherlands, water and sediment have been reintroduced in previously embanked areas in the Biesbosch, a Tidal Freshwater Wetland (TFW) in the south-west of The Netherlands. This paper presents the first results of a larger field research project examining mechanisms and controls of aggradation in the Biesbosch tidal freshwater wetland.

    The aim of this paper is to quantify both the magnitude and the spatial patterns of aggradation and erosion in one of the

formerly embanked areas, 'Kleine Noordwaard'. For this, different sources of data were combined. Sediment accumulation was determined from existing bathymetric data collected by a multibeam echosounder, existing LiDAR digital elevation models, supplemented by field observations of the thickness of the newly deposited sediment layer and field observations of the location and height of cut banks. Furthermore, we compared the rates and patterns to those in other wetlands and the sediment load of the river Rhine.





## 2   Methods

### 2.1   Study area

The Biesbosch National Park is a 9000 ha freshwater tidal wetland in the lower Rhine and Meuse delta in the southwest of
The Netherlands (see Fig. 1 (a) and (b)). The area was reclaimed in medieval times, but it became completely inundated by
5   the St. Elisabeth flood, which was a combination of two storm surges and two floods of the River Rhine between 1421 and
1424 (Zonneveld, 1959). In the subsequent two centuries, a deltaic splay developed of approximately 6 m thick (Kleinhans
et al., 2010). The lower 4 m of this splay is sand, covered by 2 metres of clay (De Bont et al., 2000). In 1861 AD, the Nieuwe
Merwede, an artificial branch of the River Rhine, was excavated through the area. As a consequence, water levels dropped and
large parts of the wetland were embanked and reclaimed as polders for agriculture during the second half of the 19th century
(De Bont et al., 2000; Bureau Noordwaard, 2006). However, since 2008, several of these polder areas have been re-opened for
river water ('depoldered') to increase the discharge capacity of the River Rhine. The study area of the Kleine Noordwaard was
among the first polder areas that have been depoldered.

The "Kleine Noordwaard" study area comprises the former Spiering, Oude Hardenhoek and Maltha polders (Fig. 1). Maps
(c) and (d) represent the surface elevation before and after depoldering of the study area. In 2008, several channels with a
width of 120 m, a maximum depth of 3 m and a side slope of 1:20 were dug throughout the area. The sandy material was
used to create islands and extra protection along the embankments (Grontmij, 2002). The original clayey polder soil remained
conserved, except in the former Spiering and Maltha polders, where the upper layer of clay had already been removed for
reinforcement of embankments. The channel in the Spiering polders in the north forms the inlet, while the channel in southern
polder Maltha forms the outlet of the system. On 7 May 2008, the embankments were opened and the inlet channel was
connected to the River Nieuwe Merwede and the outlet to the Gat van de Noorderklip. The major flow direction is from north
to south, and water and sediment are supplied by the River Nieuwe Merwede, which is a branch of the River Waal, the major
distributary of the River Rhine ( Fig. 1). Channels are always submerged, while the former polder bed comprises a system of
mud flats, which are either submerged or dry, depending on the water level. The water level is influenced by the tide, which
has a range of approximately 0.2 to 0.4 m (Rijkswaterstaat, 2016), the wind direction and speed, and the discharge of the
River Rhine. The area is composed of deep open water (18%), mud flats (31%), and a terrestrial zone (51%). In order to
reduce the hydraulic roughness, the terrestrial zone is mowed before the winter period and most of the vegetation is effectively
shortened through grazing by birds, horses, and cows. The vegetation in the terrestrial zone can be classified as dry and damp
grasslands with at the shoreline some *Mentha aquatica*, *Schoenoplectus triqueter* and *Bolboschoenus maritimus*. The mud flats
are almost bare with some pioneer species such as *Hydrodictyon reticulatum*, *Limosella aquatica*, *Veronica anagallis-aquatica*
and *Pulicaria vulgaris*. In the summer, grows locally some *Myriophyllum spicatum* in open water (De la Haye, 2011).

### 2.2   Field methods

To establish a sediment budget of the study area, we determined separate sediment budgets for the channels (surface levels
below 0.1 m above Dutch Ordnance Datum (NAP)), intertidal flats (surface levels between 0.1 and 0.5 m NAP), and the



terrestrial zone (surface levels above 0.5 m NAP). For all subareas we first determined the volumetric budgets, which were subsequently converted to mass budgets using sediment densities measured from field samples.

### 2.2.1  Channels

The change in surface elevation over multiple years was used to determine the sediment budget and the spatial pattern of
sediment accretion and loss in the channels. Surface elevation was measured by Rijkswaterstaat during consecutive bathymetric surveys in all channels in 2009, 2010, 2012 and 2015. In 2011, 2013 and 2014, additional channel sections were surveyed (see Fig. 1 (d)). Channel bed elevation was measured with a multi-beam Simrad EM3002d echosounder, combined with Netpos/LRK, and processed in QPS Quincy 8.0 (personal communication, Rijkswaterstaat). The total volumetric channel sediment budget was calculated from the difference in channel bed elevation between 2009 and 2015. Some channel sections
were dredged in 2015 and were therefore excluded from the 2015 bathymetric map. The sediment budget was corrected for this exclusion by the linear relationship between the total channel sediment budget and the channel budget without the dredged area. To analyse patterns in sedimentation and erosion over time, we used only areas from which data were collected during all surveys of 2009, 2010, 2012 and 2015. Survey data collected in the years of 2011, 2013 and 2014 were only used to assess whether changes in bed level have been consistent throughout the years.

### 2.2.2  Intertidal flats

We determined the sediment budget of the intertidal flats by measuring the vertical accretion on top of the former polder soil during field campaigns in July and October 2014. The former polder soil consists of a compact non-erodible layer of clay, which was used as marker horizon, since its colour and density are clearly distinguishable from the recently deposited sediment (sand and mud). Transparent perspex core samplers with a diameter of 59 mm and different lengths were used to
collect 126 samples of the newly deposited sediment layer in 9 transects across the central part of the Kleine Noordwaard (pink aligned area in Fig. 1). The Spiering and Maltha polders were not sampled because the non-erodible layer of clay was removed before depoldering. The thickness of the newly deposited sediment layer was measured using a ruler. In the field the texture of the newly deposited layers of sediment was classified visually as clay, sandy clay, coarse sand, fine sand, silty sand, sandy silt, or silt. Based on these texture classes, the sediment layers were classified into former polder bed (clay, sandy clay),
newly deposited sand (coarse sand, fine sand, silty sand) and newly deposited mud (sandy silt or silt). Since channels were dug and sand was replaced by depoldering of the area, we selected only samples with a base of former polder clay for further analysis. Furthermore, samples with sand on top of the base of former polder clay were corrected for the replacement of sand for depoldering. The total sediment budget of the intertidal flats was calculated by multiplying the average thickness of the recently deposited sediment layer by the total area of the intertidal flats.





### 2.2.3 Terrestrial zone

To determine the contribution of bank erosion to the budgets of the channels and intertidal flats, we measured the height and position of cut banks using a ruler and a Trimble R8 RTK GPS during field campaigns in July and October 2014. The current cut bank position and height were compared to the 2011 50 cm resolution digital elevation model of the Netherlands (AHN2),

to calculate the volume of eroded land over the period 2011-2014.

### 2.2.4 Sediment budget and trapping efficiency

The total sediment budget of the study area was calculated by taking into account the different periods for which the individual budgets of the channels, intertidal flats, and the terrestrial zone were established. To obtain the total mass of the net deposited sediment in the area, the volumetric sediment budget of the area was multiplied by the bulk density of the deposited sediment.

Bulk density of sand, deposited in the channels, was determined gravimetrically by the weight of the terrestrial sediment in a pF-ring with volume of 98.125 $cm^3$. To determine the bulk density of the mud, a 59 mm diameter Transparent Perspex core sampler was used to collect a 7 cm core with 4 cm of organic rich mud and the underlying former polder clay at the intertidal flat, because a pF-ring could not be applied to mud. The core was subsampled at a 1 cm interval in the laboratory, and bulk density was determined by the weighted average sediment particle density and the volumetric moisture content of the

samples. Moisture and organic matter content were determined from the difference in mass after oven drying at 105 °C and loss-on-ignition analysis at 550 °C, according to the standard techniques described by Heiri et al. (2001).

The trapping efficiency of the area was determined as the percentage of the load supplied to the entrance of the study area. Discharge and suspended sediment measurements at the entrance of the study area (Van der Deijl et al [in prep]) indicated that the area receives approximately 5.8 % of the total load of the River Rhine. The sediment load for the period between May

2008 and November 2015 of the River Rhine was determined from discharge measurements (10 minute interval) and daily suspended sediment concentration (SSC) at the Rijkswaterstaat Lobith gauging station near the Dutch-German border.

## 3 Results

### 3.1 Bulk density

Sediment samples indicate a bulk density of 1.47 g $cm^{-3}$ for sand and a logarithmic increase with depth from 0.75 to 1.27 g

$cm^{-3}$ for the 3 cm of mud at the intertidal flat. A maximum density of 1.5 g $cm^{-3}$ was found in the compacted former polder clay at the bottom of the core. To convert the sediment thickness to sediment mass per unit area bulk density values of 0.75, 0.94, 1.27, and 1.34 g $cm^{-3}$ were assigned to the respective 0-1, 1-2, 2-3, and > 3 cm depth intervals. For the sandy sediment that accumulated in channels or eroded from the island, we used a bulk density of 1.47 g $cm^{-3}$. The organic matter content, varied between 3.9 and 4.3 % for the former polder clay and between 2.9 and 4.3 for the mud at the intertidal flats.



## 3.2    Channels

Figure 2 shows the change in channel bed level and the yearly averaged cumulative change in channel bed volume in the central part of the area Kleine Noordwaard (pink and green areas in Fig. 1) from north to south both for the successive monitoring campaigns ((a), (b), (d) and (d)) and for the entire period ((e)). The northern part of the area (the inlet) is characterized by a negative sediment budget, due to the loss of sediment within the Spiering polders (purple in Fig. 1). The cross sectional area of the inlet has increased by erosion of the channel to the depth of the River Nieuwe Merwede. Furthermore, outer bends have eroded and bars have developed at the end of these bends. However, the channels are not able to migrate freely due to steep banks with hard bank protection, and the average width to depth ratio of the channel has decreased from 17.9 to 15.2.

Within 500 m from the inlet of the central part of the Kleine Noordwaard, the cumulative change in bed volume turns positive. Thus, the amount of sediment eroded in the Spiering polders and near the inlet of the central part of the Kleine Noordwaard is deposited in this area. Further downstream, the cumulative change in channel bed volume increases further reflecting the positive sediment budget of the Wassende Maan area (blue in Fig. 1). In contrast to the area near the inlet of the study area, the channels in the central part of the area have become shallower and width to depth ratios have increased from 20.3 to 21.9 for the through-flowing channels and from 6.4 to 6.8 for dead-ending channels.

The first outer bend of the single channel at the outlet of the study area has migrated by approximately 35-40 m in eastern direction. A steep cut bank has developed and concomitant deposition of a point bar occurred at the convex inner bend and at the end of the bend. Consequently, the second bend has migrated by approximately 20-30 m in western direction. Although the channel at the outlet of the study area has become 5-10 m wider, the width to depth ratio decreased from 22.9 to 18.3 due to channel deepening.

The temporal trend in annual average eroded and deposited sediment volumes is shown in Fig. 3 for the inlet (a), the centre (b), and the outlet (c) of the study area (represented by channel section 1; sections 2-9; and section 10 in Fig. 1). Both the annual amounts of erosion in the channels near the inlet and outlet, and the annual amounts of sedimentation in the centre of the area have decreased over time. This suggests that the channels tend to attain an equilibrium state between their geometry and the flow conditions. The decrease in the average net erosion at the inlet and outlet (black line in Fig. 3)is caused by the decrease in the average erosion rate (red in Fig. 3), since the average sedimentation rate (blue in Fig. 3) remains constant over time. The erosion rate has decreased over time because the channels reached the same depth as the River Nieuwe Merwede. The response time of the erosion (i.e the time needed to reduce the net erosion rate by 63 %) is approximately 2.2 years for the entrance, and 3.3 years for the outlet. The response time of the net sedimentation for the central part of the system is 3.1 years. The response time for net sedimentation in the entire area is 6.5 years. This value is however less accurate as it is based on only three instead of six monitoring intervals.

Although the cumulative channel sediment budget decreases from north to south due to erosion at the outlet of the area, the total channel sediment budget is positive for most periods. The total yearly sediment budget was only negative for the period between the monitoring campaigns of September 2010 and March 2011. This monitoring period includes a discharge event that occurred between 8 and 19 January 2011 with a peak river discharge of 8315 $m^3$ $s^{-1}$ at Lobith. Apparently, this event



triggered large changes in bed level during this monitoring period. The bathymetric maps in Fig. 2 show increased rates of deposition in the centre of the area during the discharge event. This increased deposition occurred at both the inner and outer bend of channels, while the bed level of the outer bend was eroded during other periods. The average change in bed level of the channels decreased from 18.5 mm year$^{-1}$ to 15.4 mm year$^{-1}$ and 12.6 mm year$^{-1}$, for the consecutive measurement intervals

2009-2010, 2010-2012 and 2012-2015. These changes in bed level correspond to an accumulation of 21.7 $10^3$ m$^3$ year$^{-1}$, 18.0 $10^3$ m$^3$ year$^{-1}$ and 14.8 $10^3$ m$^3$ year$^{-1}$ and annual average sediment budgets of 32.4 kton year$^{-1}$, 26.9 kton year$^{-1}$ and 22.1 kton year$^{-1}$, respectively. The average channel sediment budget for the entire monitoring period (2009-2015) accounts for an accumulation of 14.3 mm year$^{-1}$, which corresponds to 16.7 $10^3$ m$^3$ year$^{-1}$ and 24.9 kton year$^{-1}$.

### 3.3   Intertidal flats

Figure  4 A shows the spatial variation of the measured sediment accumulation on the intertidal flats, and Fig.  5 shows the sediment accumulation for increasing distance to the inlet of the polder area (i.e. the source of the sediment) (graph (a)); distance from the channel (graph (b)); and the height of the flats (graph (d)). Although the highest accumulation was measured along the channels, there is no significant relation between the total accumulation and the distance to the channel. This is likely partly due to the relatively high accumulation (>5 cm) at the transition from the tidal flats to the island in the centre of the

area at a distance of approximately 240-270 m from the channel. These high accumulation rates are probably caused by the redistribution of sediment eroded from the island. Although there is no significant relation between the total accumulation and the distance to the channel, field observations of the texture of the sediment layers indicated that the percentage of mud increases and the percentage of sand decreases with increasing distance to the channel. Furthermore, Fig.  5 (c). shows that the total accumulation generally decreases with increasing height of the tidal flat. This relation is significant but it explains only

9 % of the variation in accumulation. However, local variation in accumulation is large and is probably resulting from local topographic irregularities as mudflat runnels and old furrows, which are not accounted for in the digital elevation model of the area, which has a resolution of 1 m.

Between May 2008 and October 2014 an average of 43 ± 3.55 mm (standard deviation) of mud and sand, accumulated in the intertidal area. This corresponds to an accumulation rate of 6.6 ± 1.0 mm year$^{-1}$ and an annual average sediment budget of

13 $10^3$ m$^3$ year$^{-1}$ and 14.6 kton year$^{-1}$.

### 3.4   Terrestrial area

Fig.  4 B. shows all cut banks, observed in the study area in 2014. Cut banks are most abundant along the island in the centre of the system. These cut banks are located in line to those channels with a relatively long fetch for waves formed by the abundant south-westerly winds. The only cut bank formed by channel migration due to outer bend erosion, is located along the channel

in the Maltha polder. Comparison of the observed cut bank position and the 2011 digital elevation model of the area indicates that only 31 m$^3$ of the island eroded between 2011 and 2014. The sediment budget of the terrestrial area is -10.3 m$^3$ year$^{-1}$, which corresponds to an erosion of 15.4 ton year$^{-1}$.



## 3.5 Total sediment budget and trapping efficiency

Between 2008 and 2015, the total sediment budget for the 'Kleine Noordwaard' area amounted to 29.7 $10^3$ $m^3$ $year^{-1}$, which corresponds to a net area-averaged sedimentation rate of 5.1 mm $year^{-1}$ and an import of 39.5 kton $year^{-1}$ for the first 6.5 years after depoldering. Sedimentation of 16.7 $10^3$ $m^3$ $year^{-1}$ in the channels accounts for approximately 60 percent of the total

budget. The vast majority of the remaining 40 % comprises the annual sedimentation of 13.0 $10^3$ $m^3$ $year^{-1}$ in the intertidal area. Remobilization of sediment by erosion of cut banks occurred at a negligible rate of about -10.3 $m^3$ $year^{-1}$. During the first 6.5 years after depoldering of the area approximately 46 % of the incoming sediment in the study area and approximately 3 % of the incoming sediment at Lobith (the Dutch-German border) was trapped by the area. Consecutive measurements of channel bathymetry indicate that aggradation has occurred in all years, but the actual channel aggradation rate has decreased

over the years (see Fig. 3) . Assuming a constant accumulation and erosion rate for the intertidal- and terrestrial zone, we have accounted for the decrease in channel aggradation over the years by calculating the prospected sediment budget for the coming years using the bathymetric measurements of 2012 and 2015. The prospected sediment budget of the study area amounts 25.5 $10^3$ $m^3$ $year^{-1}$, which corresponds to a net area-averaged sedimentation rate of 4.4 mm $year^{-1}$.

## 4 Discussion

### 4.1 Patterns in sedimentation

Most previous studies on patterns of sedimentation and erosion in tidal wetlands either focussed on marshes, tidal flats or tidal channels. Yet the present study was designed to determine both the volume and the spatial patterns in sedimentation and erosion of the entire Kleine Noordwaard tidal freshwater wetland, which includes terrestrial area, tidal flats, and channels. The largest sedimentation rates in the Kleine Noordwaard were found in the channels. This is in agreement with the findings by

Siobhan Fennessy et al. (1994); Anderson and Mitsch (2006); Mitsch et al. (2014) who showed that in deep open water areas, more sediment accumulates than in shallow open water areas that are more easily subjected to resuspension by wind-driven and biological sediment disturbances.

Consecutive bathymetric measurements showed that the channels tend to attain an equilibrium state between their geometry and the flow conditions. Channel bed erosion and an associated decrease in width to depth ratio took mainly place near the

inlet and outlet of the system, where only one channel is present. Sediment accumulation at the bed and an increase in the width to depth ratio occurred in the centre of the system, due to the increased cross-sectional area and associated decreased flow velocities. Application of the hydraulic geometry relations of Klaassen and Vermeer (1988) and the Engelund and Hansen predictor (1967) for the total sediment transport capacity in channels, as described by Marra et al. (2014), indicates that the transport capacity of the two main channels in the centre of the area is approximately 46 % of the capacity of the single

channel near the inlet and outlet of the system. When it is assumed that both the inlet and centre of the system have reached their equilibrium state with the flow conditions, the negative sediment budget of the inlet for the period 2012-2015 can be seen as the total maximum transport capacity of this channel. The channels in the centre of the system have a relative transport





capacity of only 46 %, so 54 % of the incoming material is deposited by the reduced transport capacity of the two parallel channel systems. However, the reduced transport capacity of the channels explains only 24 % of the positive sediment budget in the centre of the system, which indicates that not only the bifurcation, but also the presence of the wide and shallow intertidal area, results in enhanced sedimentation in the centre of the area.

Observations of freshly deposited material show a trend of declining sand accumulation and increasing mud accumulation away from the channels, which is in line with previous studies of Neubauer et al. (2002); Reed et al. (1999); Temmerman et al. (2003); French and Spencer (1993); Siobhan Fennessy et al. (1994) and Anderson and Mitsch (2007). Furthermore, a significant negative trend between accumulation and the height of the location was found, which has also been reported by French and Spencer (1993); Middelkoop and van der Perk (1998); Reed et al. (1999); Temmerman et al. (2003); Thonon et al.

(2007). Although the height and distance from the channel do control sediment accumulation, their influence is not as strong as observed in most marshes or river floodplains. There are three likely causes for the weak relations we found in the Kleine Noordwaard study area: 1) the small gradient in surface topography and a large variation in micro-relief by the presence of mudflat runnels, old furrows, and ditches (Whitehouse et al., 2000; Takekawa et al., 2010; Fagherazzi and Mariotti, 2012); 2) the absence of vegetation (Neubauer et al., 2002; Darke and Megonigal, 2003; Vandenbruwaene et al., 2015), and 3) the small

tidal range and water depths (Mariotti and Fagherazzi, 2013). These three factors result in an uniform sediment distribution by firstly, micro topographic flow paths during low water levels (Hupp et al., 2008; Temmerman et al., 2005a); secondly, sheet flow during high water levels (Vandenbruwaene et al., 2015; Temmerman et al., 2005a); or thirdly, a relatively large impact of shear stress (Fagherazzi and Mariotti, 2012) of wind waves and currents, which hamper sediment settling and/or promote sediment redistribution across the tidal flats. However, it could also be argued that the absence of a clear relation between the

total mud accumulation and the distance to the channel is due to the fact that sedimentation is not limited by sediment depletion from the flow over the intertidal flats. This would imply that the water and sediment remain well-mixed across the intertidal flats or that the residence time of water above the flats is relatively short.

In contrast to studies of van de Koppel et al. (2005); Fagherazzi and Wiberg (2009); Tonelli et al. (2010); van Proosdij et al. (2006), cut bank retreat does not significantly contribute to the total sediment budget of the Kleine Noordwaard study area,

where except for one cut bank that was formed by channel migration, all cut banks are formed by wind wave erosion. Wind waves are generated by the transfer of energy from wind to the water surface. This transfer is determined by the length of the fetch (the unobstructed water surface over which the wind blows), and the mean water depth over this fetch (Fagherazzi and Wiberg, 2009). A fetch of 400 m and an average water depth of 0.2 m are typical for the study area. According to the approach of Fagherazzi and Wiberg (2009), which estimates the maximum wave height and accompanying bed shear stress, wind waves

do not exceed a height of 8 cm for wind speeds up to 15 m s$^{-1}$ under these conditions of water depth and fetch length. This is probably the major reason for the low rates of cut bank retreat in the majority of the study area. In accordance with Leonardi et al. (2016) who have shown that bank deterioration is linearly related to wave energy the low rate of cut bank retreat in the study area can be attributed to the low wave height. Higher wind waves and an inherent larger rate of cut bank retreat can only occur at the boundary of the terrestrial zone in the northeast of the area and at the southwest edge of the island. Both locations



are exposed perpendicular to the southwest oriented channels, which are relatively deep and have a long fetch for the most abundant southwesterly winds.

In spite of their low height wind waves cause erosion on tidal flats when the wave-generated shear stress exceeds the critical bed shear stress for erosion. A critical bed shear stress of 0.35 N m$^{-2}$ (Mitchener and Torfs, 1996) for sand is only exceeded for

typical conditions of fetch and water depth and very high wind speeds >13 m s$^{-1}$. However, resuspension of unconsolidated mud with a critical shear stress of 0.05 N m$^{-2}$ (Mitchener and Torfs, 1996), already takes place at a wind speed greater than 5.5 m s$^{-1}$, which was the case during approximately 20 % of the total study period. This suggests that resuspension of the newly deposited material at the intertidal flats takes place regularly. The results from this study do not allow drawing conclusions about the fate of the resuspended sediment, but possibly part of this sediment is exported from the study area during strong

wind events or high river discharges.

## 4.2 Sediment budget

The results of this study indicate that the area Kleine Noordwaard functions as a sediment trap. Both the net area-averaged aggradation rate of 5.1 mm year$^{-1}$ for the period since the opening of the polder area and the estimated actual net area-averaged aggradation rate of 4.4 mm year$^{-1}$ are well within the reported ranges for accumulation rates on floodplains, wetlands, fresh-

and salt-water marshes (see Table 1). Furthermore, the aggradation rate in the Kleine Noordwaard is within the range of the mean overbank sedimentation rates over the last century (0.18-11.55 mm year$^{-1}$ with a mean of 2.78 mm year$^{-1}$) on the upstream river floodplains along the River Waal as reconstructed by Middelkoop (1997) from heavy metal profiles in floodplain soils of the River Rhine. However, the aggradation is larger than the 1-2 mm year$^{-1}$, reported by Bleuten et al. (2009) for the Mariapolder, a re-opened polder area located north of the River Nieuwe Merwede, close to the Kleine Noordwaard. This is

likely due to the fact that the Kleine Noordwaard study area receives a larger supply of water and sediment from the River Nieuwe Merwede than the Mariapolder, that has only a single inlet/outlet and is only subject to tidal in- and outflow.

The consecutive measurements of channel bathymetry indicate that aggradation is consistent, but that the actual aggradation rate varies over the years. Although the largest rates of erosion and sedimentation occurred during the 2011 peak discharge event of the river Rhine, the total erosion and sedimentation rates have decreased to 37 % of their initial value within 2 to 3.5

years. This is confirmed by the slightly lower net sediment accumulation rate of 4.2 mm year$^{-1}$, for the period 2014-2015, as determined from the field measurements of water levels, flow velocities, and suspended sediment concentrations at both the in- and outlet of the study area (Van der Deijl et al., in prep.). The trend of a decrease in erosion and sedimentation rates in newly created wetlands was also found by Vandenbruwaene et al. (2012), who found in a newly created wetland a decrease in the net channel erosion and sedimentation rates towards zero after 4 years.

## 30 4.3 Implications for management

The findings of this study have a number of practical implications for future river delta restoration. The current net area-averaged aggradation rate of 4.4 mm year$^{-1}$ is just enough to compensate the actual rate of sea-level rise and soil subsidence in the Netherlands, which are 2 mm year$^{-1}$ (Ligvoet et al., 2015) and 0.5-2.5 mm year$^{-1}$ respectively for the Biesbosch (Kooi et al.,



1998). However, sedimentation in the area can not compensate the high end scenarios for sea-level rise of 0.4 to 10.5 mm year$^{-1}$ as calculated for the Netherlands by Katsman et al. (2011), especially since freshly deposited sediment will compact over the years and thus result in a lower net accumulation. The area has only trapped approximately 46 % of the incoming sediment in the study area and approximately 3 % of the incoming sediment at Lobith (the Dutch-German border), which indicates that

it is possible to optimise the sediment trapping efficiency and increase the aggradation rate of the study area. This might be achieved by reducing the resuspension at the intertidal flats by a decrease in the wave-generated shear stresses. This could be achieved by decreasing the fetch by the establishment of vegetation or by the construction of topographic irregularities. The aggradation rate in the study area could be enhanced by increasing the supply of water and sediment to the area, for example by modifying the inlet geometry. Such measures to increase the sediment trapping efficiency and aggradation rates can also be

applied in other wetlands, especially those where current aggradation rates are not sufficient to compensate sea-level rise and soil subsidence.

## 5    Conclusions

Existing data sets and field data were used to quantify both the amount and the spatial patterns in sediment accumulation and erosion in the formerly embanked area Kleine Noordwaard. The main conclusions of this study are:

1. During the period 2008-2015 the total sediment budget of the 'Kleine Noordwaard' study area amounted to 29.7 $10^3$ m$^3$ year$^{-1}$, which corresponds to a net area-averaged aggradation rate of 5.1 mm year$^{-1}$ and an import of 0.95 kton year$^{-1}$. The area has trapped approximately 46 % of the sediment delivered to the study area, which is approximately 3 % of the incoming sediment at Lobith (Dutch-German border).

2. Largest rates of accumulation (14.3 mm year$^{-1}$) occurs in the channels, which accounts for approximately 60 percent of

the total budget, the other 40 percent comprises sedimentation in the intertidal area at an average rate of 6.6 mm year$^{-1}$. Cut bank retreat does not significantly contribute to the total sediment budget.

3. The rates of accumulation and erosion in the channels decrease over time, which imply the channels tend to equilibrium between the flow conditions and their geometry.

4. At the intertidal flats sand aggradation occurs primarily at low lying locations close to the channels, while mud aggrada-

tion occurs primarily further away from the channels and inlet.

5. The current aggradation rate is enough to compensate for the actual rate of sea-level rise and soil subsidence, but not for the high end scenarios of sea-level rise. The aggradation rate of the Kleine Noordwaard study areas and wetlands in general could be enhanced by optimizing the sediment trapping efficiency of the area or by increasing the supply of water and sediment to the area.

*Data availability.* We will make the following data available via the Dryad repository (http://datadryad.org/):



1. Difference in channel bed level for each monitoring period (TIF) (Figure 2, Figure 4)

2. Initial Digital Elevation Model of the study area (TIF) (Figure 2, Figure 4)

3. Locations of inlet centre, and outlet (TIF)

4. Location of dredged area (TIF)

5   5. Total accumulation at the intertidal flats (ascii xyz) (Figure 4a)

6. Location and height of the cutbanks (ascii xyz) (Figure 4b)

*Author contributions.* H. Middelkoop and M. van der Perk have designed the research proposal. E.C. van der Deijl collected and analysed the data. E.C. van der Deijl, M. van der Perk and H. Middelkoop interpreted the data and finally E.C. van der Deijl prepared the manuscript with the contributions, revisions and final approval from all co-authors.

10   *Competing interests.* The authors declare that they have no conflict of interest.

*Acknowledgements.* This project is financed by the Dutch Technology Foundation STW (project nr. 12431). We thank Staatsbosbeheer, Rijkswaterstaat, Dr. Hans de Boois, Eelco Verschelling, Dr. Wim Hoek, Renske Visser, Nanda Kik and Wouter Zonneveld for the provided data, assistance, logistic support and knowledge.



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



Table 1: Sedimentation rates in various types of delta compartments

| Type | Accumulation [mm year⁻¹] | Accumulation | Turbidity | Amplitude [m] | Method | Source |
|---|---|---|---|---|---|---|
| Marshes | 0.6 | | | | Cores (137Cs , 210 Pb) | Craft and Casey (2000) |
| Floodplains | 1.1 | | | | Cores (137Cs , 210 Pb) | Craft and Casey (2000) |
| Wetland | 1.2 | | | | Cores (137Cs , 210 Pb) | Craft and Casey (2000) |
| Wetland | *1.5* | 1.1 - 8.4 g m⁻² day⁻¹ | | | Sediment traps (filters) | Reed et al. (1997) |
| TFW 1.5 | | | | 0.4 - 0.5 | Waterlevels, watersamples | Bleuten et al. (2009) |
| Floodplains | 2.2 | 2.56 kg m⁻² year⁻¹ | 30 mg l⁻¹ | | Sediment traps (artificial grass) | Middelkoop and Asselman (1998) |
| Floodplains | 2.8 | | 30 mg l⁻¹ | | Cores | Middelkoop (1997) |
| Floodplains | 2.5 | | | | Cores (137Cs , 210 Pb) | Gell et al. (2009) |
| Crevasse splay | 2.5-3 | | | | Cores (137Cs , 210 Pb) | Day et al. (2016) |
| Salt Marshes | 3.9 | | 50-150 mg l⁻¹ | 3.2 (nt) - 6.4 (st) | Marker horizons, sediment traps | French and Spencer (1993) |
| 2 River Marshes | *4.1* | 4. - 4.9 kg m⁻² year⁻¹ | 17 - 35 NTU | | Marker horizons | Anderson and Mitsch (2006) |
| TFW | 4.9 | | | 0.7 (nt) - 0.9 (st) | Sediment traps (tiles), cores (Be7) | Neubauer et al. (2002) |
| 3 Salt Marshes | 0 - 11 | | | 1.3 - 2.3 | Marker horizons, surface elevation, clay layer thickness | van Wijnen and Bakker (2001) |
| Floodplains | 6 | | 90 (mean) - 400 mg l⁻¹ (flood) | | Sediment traps and optical backscatter sensors | Hung et al. (2014a, b) |
| Salt Marshes | 6 | | 50 mg l⁻¹ | 3.2 (nt) - 6.4 (st) | Sediment traps (filters) | French & Spencer, unpubl. in Reed et al. (1999) |
| Wetland Basin | 5 - 10 | | | | Sediment traps (bottles) | Siobhan Fennessy et al. (1994) |
| TFW | 8.7 | 1 g cm⁻² year⁻¹ | | 3.2 (nt) - 6.4 (st) | Sediment traps (tiles) | Pasternack and Brush (2001) |
| Salt Marsh | 8 - 14 | | 300 mg l⁻¹ | 14 | DEM, Sediment plates | van Proosdij et al. (2006) |



| Type | Accumulation [mm year⁻¹] | Accumulation | Turbidity | Amplitude [m] | Method | Source |
|---|---|---|---|---|---|---|
| TFW | 12 | | 9 mg l⁻¹ | 1.8 (st) | Sediment traps (tiles), Marker horizons | Darke and Megonigal (2003) |
| TFW | 12 | | 5 mg l⁻¹ | | Marker horizons, Surface elevation tables | Calvo-Cubero et al. (2013) |
| Wetland | 14 (13 years), 42 (1996), 55 (2009) | | | | Sediment traps (bottles), Marker horizons | Mitsch et al. (2014) |
| TFW | 16 | | 27 mg l⁻¹ | | Marker horizons, Surface elevation tables | Calvo-Cubero et al. (2013) |
| 14 Salt Marshes | 0.02 - 17 for marshes of 500 - 10 years old | | | 2-3 (nt) - 5-6 (st) | Model | Pethick (1981) |
| salt marsh | 12.3 | 13.8 - 63.7 g m⁻² day⁻¹ | 10-42 mg l⁻¹ | 1.8 (st) | Sediment traps (filters) | Leonard (1997) |
| Salt marsh | 21.3 | 0.2 - 67 g m⁻² tide⁻¹ | 300 mg l⁻¹ | 14 | Sediment traps (filters) | van Proosdij et al. (2006) |
| Salt Marsh | 22 | | 130 mg l⁻¹ | 11 | Sediment traps, geotracer gps | van Proosdij et al, 1999 in Davidson-Arnott et al. (2002) |
| TFW | 27 | | 21 mg l⁻¹ | 1.8 (st) | Sediment traps (tiles), Marker horizons | Darke and Megonigal (2003) |
| River channel bed | 39 | | 1037 million tonnes year⁻¹ | | Gauging stations | Islam et al. (1999) |
| TFW | 180 | | | | Fast static gps survey | Auerbach et al. (2015) |



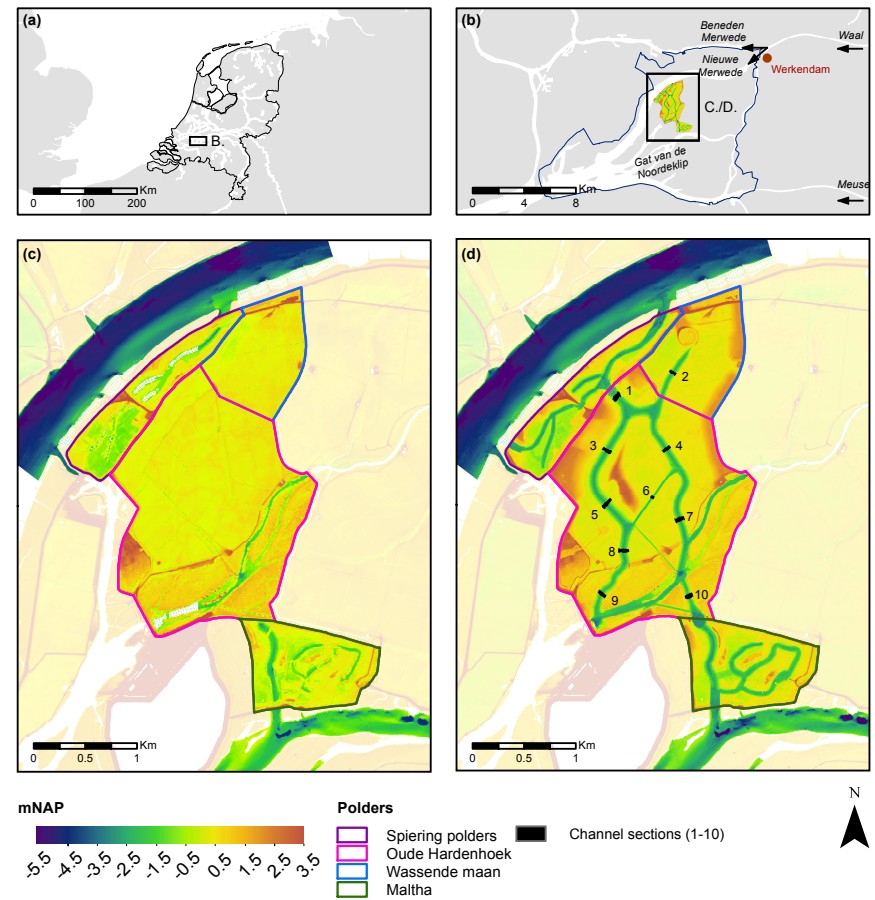

**Figure 1.** The study area Kleine Noordwaard, which is located within the Biesbosch Freshwater Tidal Wetland, in the lower Rhine and Meuse delta in the southwest of the Netherlands (a and b). Elevation is shown in meters, with respect to the Dutch Ordnance Datum (NAP) for the period before (c) and after depoldering (d).





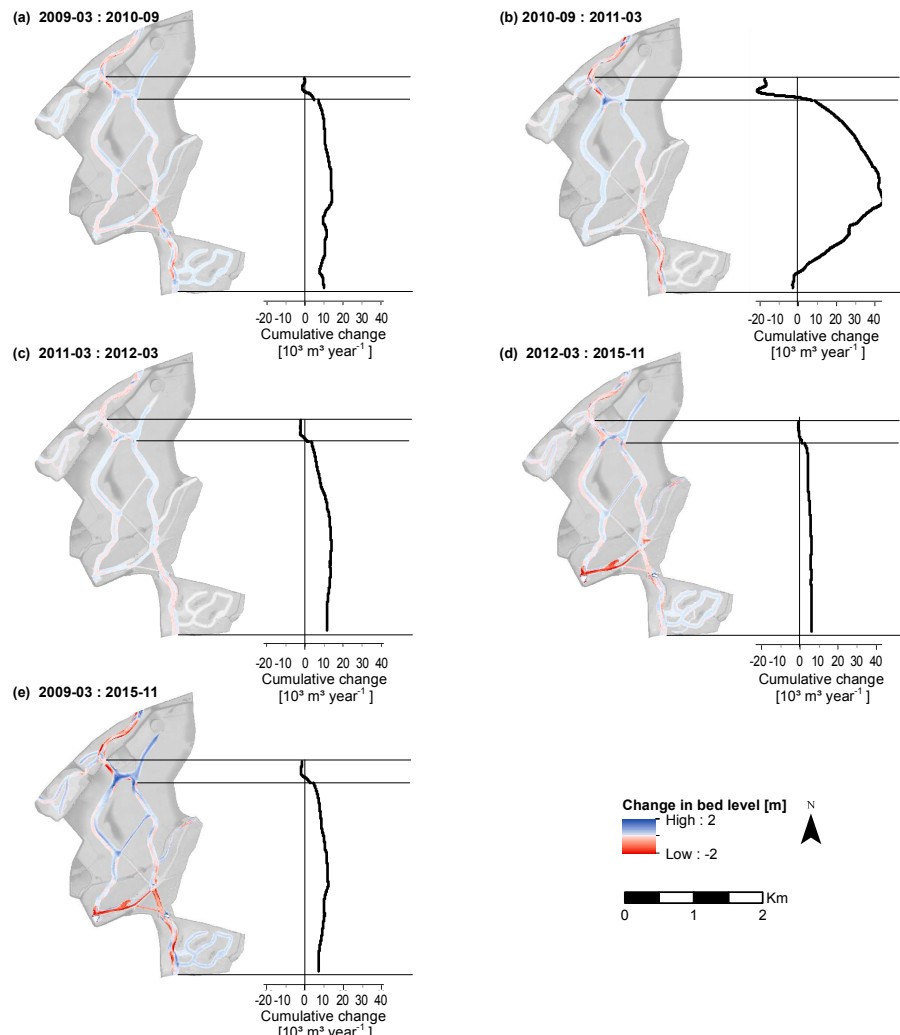

**Figure 2.** The difference in channel bed level and the cumulative channel bed volume are shown for each monitoring period. The cumulative channel bed volume is shown along a N-S transect starting from the Spiering polders (purple in Fig. 1). The budget of the Wassende Maan (blue in Fig. 1) is added at once at the second black line





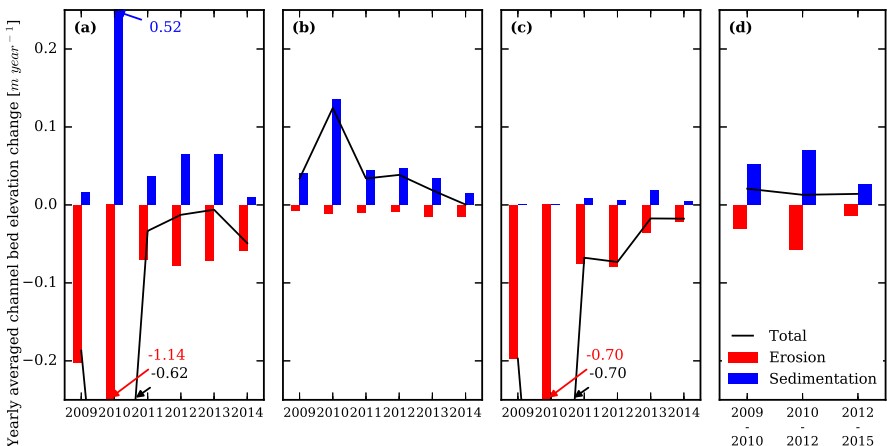

**Figure 3.** The yearly averaged channel section deposition (blue bars), erosion (red bars) and total change in height (black lines) for the inlet (a), the centre (b), the outlet (c) and the entire study area (d), which was opened 7 May 2008 and experienced a peak discharge event from 8 to 19 January, 2011. Yearly averaged channel section deposition erosion and total change in height deposition rates for the entire area are only available for the periods 2009-2010, 2010-2012 and 2012-2015, since bathymetric measurements were only executed in all channels in 2009, 2010, 2012 and 2015; and in channel sections (see Fig. 1 (d)) in 2011, 2013 and 2014.




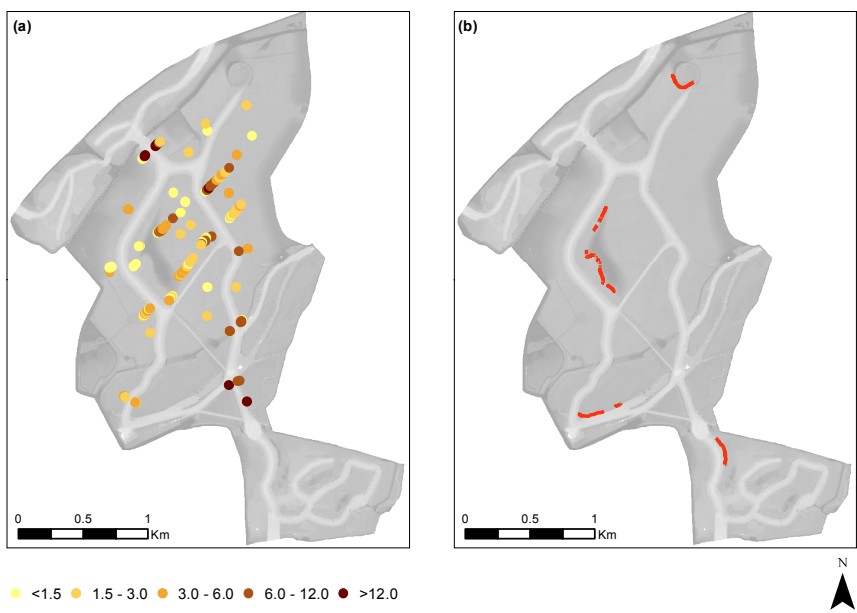

**Figure 4.** The total accumulation [cm] at the intertidal flats (a) and the location of the cutbanks (b) as measured during field campaigns of 2014



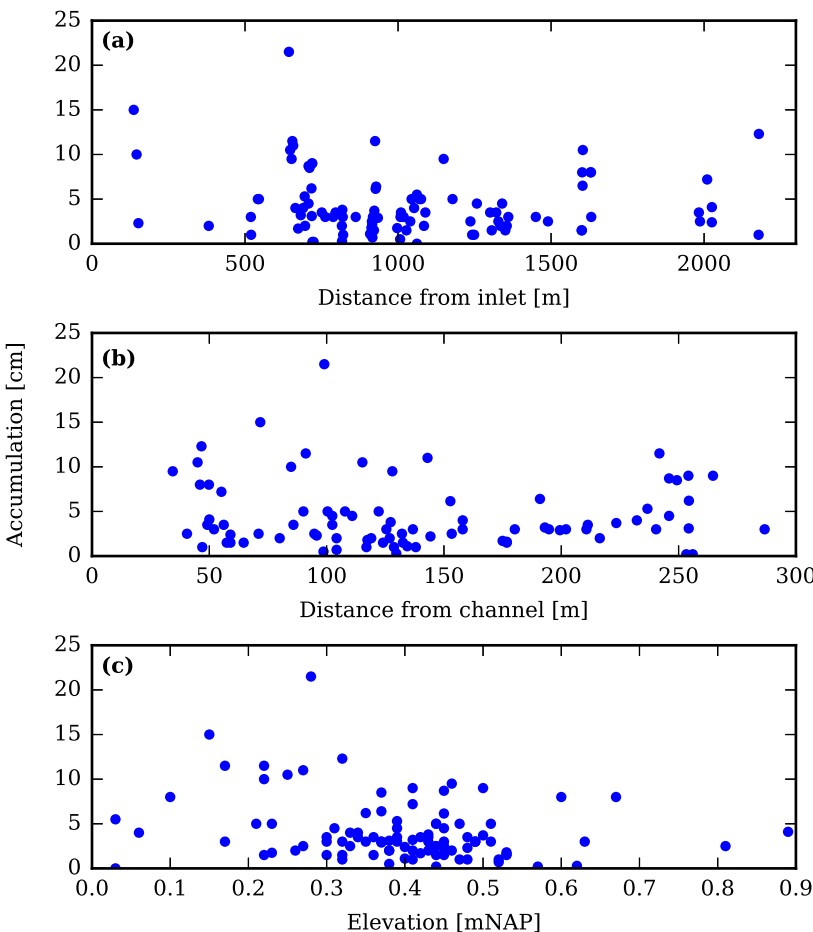

**Figure 5.** Total accumulation during the period 2008-2014 at the intertidal flats, with respect to the distance from the inlet of the system (a), distance from the channel (b), and the elevation of the flats (c)