# Peer review of "Establishing a sediment budget in the newly created 'Kleine Noordwaard' wetland area in the Rhine-Meuse delta"

_Earth Surface Dynamics, 2017_

## Referee Comment (RC1) · Anonymous Referee #1 · 8 Jun 2017

Interesting paper, describing the evolution of a restoration project based on a nice data set.

Remarks p2 line 11: sedimentation rates: this is not the same as accumulation rate or aggradation. The latter is the net elevation change (= sedimentation minus compaction or subsidence). These terms are mixed in the paper. E.g. in table 1, it is not clear if aggradation or sedimentation is shown. The results of this paper show aggradation data: change in elevation based on bathymetric surveys in the channels, both also sedimentation data based on cores on the flats. This difference is not properly discussed. This sedimentation rate might be higher than the elevation change due to subsidence.

[Figure]

Table 1 show in the header "Turbidity", while often suspended matter concentrations are shown.

p2 line 14: the authors describe that sedimentation is controlled by frequency and duration of inundation, SPM concentration in the feeding channel. However, none of these data are shown for the study site! The latter would be very interesting to show, especially because table 1 shows this for other sites.

p3: description of the study site is not accurate enough. It is described as a tidal wetland. I interpret this as a wetland where water flows in and out, twice a day, based on the tidal cycle. The site has however an inlet upstream and an outlet downstream. Is there no change in current direction, causing inflow during flood at the downstream opening? In a tidal wetland, I would expect channels, bare flats and tidal marshes. Are there no marshes? The terrestrial zone is mowed to reduce hydraulic roughness: this indicated the area can be flooded. No information on this flooding (frequency, height, duration) is given.

p4 line 28: there seems to be a big heterogeneity in the thickness of the deposited sediments. Using an average is probably very inaccurate to calculate the total sediment budget. Why not using a model, using elevation. The authors describe that sedimentation is significantly correlated with elevation.

p5: Terrestrial zone: only erosion is described here. But from the introduction, I derive that this part floods occasionally? Given the large surface of this part (>50% of the total area), even a very small sedimentation during winter can have a significant effect on the total budget.

p5 line 17. No information on how the incoming load was calculated, is given. This is however important information. For the River Rhine some info is given: 10 minute intervals for discharge, daily SSC concentrations, between 2008 and 2015. Is this done with the same accuracy for the inflow in the area? Is there no inflow at the downstream opening during flood (see previous remarks)? Measurements of SSC at inflow and

outflow would be a good way to calculate a trapping efficiency. The paper often refers to the total load of the river Rhine at the German border. How relevant is this? It is so far away from the site, after many branches and tributary rivers. I am more interested in knowing the total load of the Nieuwe Merwede that feeds the site with water and sediments: SSC and discharge in this river and SSC and discharge entering the study site. Unfortunately this information is not given. P8 line 7: how is this 46% calculated? P10 line 22: is there seasonality in the sedimentation? What is the seasonality in discharge, in SSC, in tidal amplitude? How much do peak events contribute to the budget? The peak in 2011, was it only a peak in discharge or did this event also had higher SSC?

---

## Referee Comment (RC2) · Anonymous Referee #2 · 14 Jul 2017

The authors present results of a comprehensive study on the morphological development of a recently opened wetland, based on an extensive data set.

First of all, I completely agree with the questions raised by Reviewer #1.

The text cites a paper as to be in preparation, by the authors, which is not in the list of references. That, and conference abstracts by the authors, suggest the availability of potentially more data on SSC and current velocities at the entrance to the wetland, which are, in here, deeply missed. I guess, there is a good reason for the suspected splitting of the data set.

So, since no more information is presented here on the forcing, tide-driven dynamics,

etc., the focus should be on the analysis of geospatial information. In this, I think the data source is not (yet) presented to the reader in a proper way. From the suggested high quality of MBES and LIDAR data, I had expected simply better plots, e.g., from one flat in the center of the wetland and surrounding channels, showing more details of aggradation and erosion patters.

The color scale in Fig. 1 doesn't help. The interesting elevation range, -2 m to 1 m, is essentially not resolved.

There are quite some simple methods around, based on gridded spatial data, e.g., the maximum bed elevation range to differentiate between more and less active regions, or, vertical dynamic trend analysis to show - what is already in the text - stagnation of morphological change with time at specific locations. The DEMs could have also been used to extract cross-sections of channels, or longitudinal transects from some channel thalweg, all means to give the reader a good impression of morphological changes. I'm just suggesting. How to proceed, and what to change, depends on the focus.

Tja, what is the actual focus? The determination of the budget? Analysis of transport both into and inside the area? Geomorphological changes of flats, banks, channels? Analysis of the sedimentology, including transport of the coarse and cohesive fractions?

I find the focus unclear and the Introduction and Discussion symptomatically unspecific, e.g., page 9 line 24: "cut bank retreat does not significantly contribute to the total sediment budget". If the paper was about the budget, why is the statement, cut banks would not contribute to that budget, followed by a detailed comparison of cut bank dynamics with literature? That is a question of the focus. The same could be said regarding sand deposition in the deeper channels.

The discussion essentially remains on the level of a comparison. In each part of the discussion, process based considerations are used to explain the situation at the specific location. Which is cool. But I asked myself at the end of each paragraph, what

could be concluded from this discussion for similar geomorphological settings? I had the impression, and maybe I'm wrong, that the discussion did not lead to the conclusions.

The Conclusion is essentially a summary of Results.

On one hand, it is said that the wetland is a trap for sediments. On the other hand, it is deduced at least for the channels that the domain is approaching some kind of equilibrium, which certainly didn't exist from the beginning. So, trapping occurs only because of the non-equilibrium state at the opening of the wetland? Which grain sizes could actually be trapped? Is it realistic that sand is transported into the wetland? If trapping was the goal (Paola et al., 2011, is cited in the Introduction), should we conclude that, to efficiently trap sediments, we would need to open new wetlands every once in a while? Ok, seriously, I find this important, as, e.g., around many estuaries we are having this discussion right now, everywhere in Europe.

So, in summary, I hope for a shorter text, with a clear focus. The plots should reflect the data quality. The Discussion should lead to the Conclusions.

Minor comments:

P6 L7 An interesting aspect is that "channels cannot migrate freely". Any ideas what this could mean for small-scale morphodynamics and the distribution of sediments? What is that "bank protection"?

P6 L23 Any particular reason, why the term "morphodynamic equilibrium" is persistently avoided, using "equilibrium state between their geometry and the flow conditions", instead?

P4 L14 "whether changes . . . had been . . . years"

P8 L17 "Yet . . . channels" example of many sentences which can simply be deleted without changing anything in the text

P9 L25 "Wind waves . . . fetch" common knowledge, not required

P10 L3 Entire paragraph: example for text with too many assumptions, for my taste, since in the end this does "not allow drawing conclusions"

---

## Author Comment (AC1) · 21 Sep 2017

We thank the reviewers for taking the time to read and review our manuscript. We address each specific comment below:

**Reviewer comments are bold**
Author comments are in plain text
The changes in manuscript can be found in the manuscript with track changes

[Figure]

**1 Anonymous Referee #1**

**Remarks p2 line 11: sedimentation rates: this is not the same as accumulation rate or aggradation. The latter is the net elevation change (= sedimentation minus compaction or subsidence). These terms are mixed in the paper. E.g. in table 1, it is not clear if aggradation or sedimentation is shown. The results of this paper show aggradation data: change in elevation based on bathymetric surveys in the channels, both also sedimentation data based on cores on the flats. This difference is not properly discussed. This sedimentation rate might be higher than the elevation change due to subsidence. Table 1 show in the header "Turbidity", while often suspended matter concentrations are shown.**

We agree with the reviewer that the sedimentation rates presented may also be affected by subsidence and compaction. The elevation change based on the bathymetric surveys includes both subsidence and compaction. According to Kooij et al., 1998 the rate of subsidence (deep vertical regional land movement) in the Biesbosch area is low and amounts to -0.25-0 mm year-1, including compaction (-0.1-0 mm year-1mm year-1), isostasy (-0.1 mm year-1), and tectonics (-0.03 mm year-1). However, shallow surface compaction is likely considerable, but this has affected both the change in elevation based on the bathymetric surveys and the sedimentation data based on the cores on the flats. Therefore, we have decided to use the term net sedimentation for the rate of change in height based on the bathymetric data. This is explained in line 32 on p. 2. We use the terms sediment deposition or sedimentation for the average amount of sediment deposited annually derived from the cores on the flats.

Furthermore we have made the following changes in table 1:

- We have adapted the title of the second column from Accumulation [mm year-1] to net sedimentation [mm year-1] .

- We have changed the title of the third column from Accumulation to Deposition

- We have changed the title of the fourth column from Turbidity to SSC / Turbidity

- We have added a new column in between the former second and third columns. This column has the title Deposition [mm year-1]. When a source only gives the deposition in volume, the concomitant change in height, calculated using a sediment density of 1150 kg m-3, is shown in italics

- The caption of the table has been adapted to "Net sedimentation and deposition in various types of delta compartments. When a source only gives the deposition in volume, the concomitant change in height, calculated using a sediment density of 1150 kg m-3, is shown in italics"

- we have combined the studies of van Proosdij et al. 2006 and 2006a in Allen Creeck in one row.

- we have added the deposition rates for the Gleason and Walkerton marshes in the study of Darke and Megonigal 2003, for which only the accumulation rates were given in the first version of the manuscript

- we have added the deposition rates for the study of Bleuten et a. 2009

**p2 line 14: the authors describe that sedimentation is controlled by frequency and duration of inundation, SPM concentration in the feeding channel. However, none of these data are shown for the study site! The latter would be very interesting to show, especially because table 1 shows this for other sites.**

We have addressed this comment by an expansion of the description of the study site

**ESurfD**
[Figure]

in section 2.1. The exact changes in this section will be illustrated in our reply to the next comment

**p3: description of the study site is not accurate enough. It is described as a tidal wetland. I interpret this as a wetland where water flows in and out, twice a day, based on the tidal cycle. The site has however an inlet upstream and an outlet downstream. Is there no change in current direction, causing inflow during flood at the downstream opening? In a tidal wetland, I would expect channels, bare flats and tidal marshes. Are there no marshes? The terrestrial zone is mowed to reduce hydraulic roughness: this indicated the area can be flooded. No information on this flooding (frequency, height, duration) is given..**

In section 2.1, we have added a short description of the water levels, discharge, and suspended sediment concentration in the channels of the study area. Furthermore, based on the digital elevation model and water level measurements by Rijkswaterstaat (available for the entire period since opening of the study area), we changed the subdivision of the study area from channels, tidal flats, and terrestrial zone to subtidal area (surface elevation <0.125 mNAP, always submerged) , flats (0.125 mNAP - mean sea level), low marshes (mean sea level - mean high water), high marshes (mean high water - extreme high water), and a terrestrial zone (> extreme high water). We have also added information on the flooding frequencies of these different areas.

To clarify both the morphology of the study area and the division of the study area in subtidal, flats, marshes and a terrestrial zone, we have added a new figure (Fig2), which includes a cross section of the study area. The location of this cross section is also indicated in Fig1.

We have also added a more detailed description of the discharge regime and suspended sediment concentrations of the River Rhine. Unfortunately, no measurements are available for the River Nieuwe Merwede, which is the feeding channel of the study area. The exact inflow of water and sediment from the Rhine into the Nieuwe Merwede is unknown, since it varies due to the tide, the river discharge of the River Rhine, and the artificially controlled discharge through the gates of the downstream located Haringvliet barrier into the North Sea.

**p4 line 28: there seems to be a big heterogeneity in the thickness of the deposited sediments. Using an average is probably very inaccurate to calculate the total sediment budget. Why not using a model, using elevation. The authors describe that sedimentation is significantly correlated with elevation.**

We thank the reviewer for this suggestion. We now estimate the sediment budget using a model taking into account the surface elevation. For this, we divided the study area into navigable channels, subtidal area and intertidal area (flats, low and high marshes) and a terrestrial zone (not flooded during the study period). The sediment budget for the navigable channels was calculated from the bathymetric maps. For the subtidal area, the sediment budget was calculated using the average sedimentation in this area. The sediment budget for the intertidal area was calculated using a negative exponential relation between sediment deposition and elevation.

The exact changes in the description of the methods and results, can be found in the manuscript with track changes

**p5: Terrestrial zone: only erosion is described here. But from the introduction, I derive that this part floods occasionally? Given the large surface of this part**

**(>50% of the total area), even a very small sedimentation during winter can have a significant effect on the total budget.**

In the new model-based estimation of the sediment budget (see reply to previous comment), we now include the entire intertidal area, which is flooded occasionally. The new net sedimentation rates, and trapping efficiency are indeed higher than the estimates in the previous version of the manuscript.

The exact changes in the description of the methods and results, can be found in the manuscript with track changes

**p5 line 17. No information on how the incoming load was calculated, is given. This is however important information. For the River Rhine some info is given: 10 minute intervals for discharge, daily SSC concentrations, between 2008 and 2015. Is this done with the same accuracy for the inflow in the area? Is there no inflow at the downstream opening during flood (see previous remarks)? Measurements of SSC at inflow and outflow would be a good way to calculate a trapping efficiency. The paper often refers to the total load of the River Rhine at the German border. How relevant is this? It is so far away from the site, after many branches and tributary rivers. I am more interested in knowing the total load of the Nieuwe Merwede that feeds the site with water and sediments: SSC and discharge in this river and SSC and discharge entering the study site. Unfortunately this information is not given.**

We have added a more detailed description of the discharge regime and suspended sediment concentrations of the River Rhine in the description of the study area. Unfortunately, there is no measurement station in the Nieuwe Merwede, which is the feeding channel of the study area. The exact division of water and sediment from the River

Rhine towards the River Nieuwe Merwede is unknown, since it varies due to the tide, the river discharge of the River Rhine, and the artificially controlled discharge through the gates of the downstream located Haringvliet barrier into the North Sea. However, we measured water levels, flow velocities and suspended sediment concentrations at a 10 minute interval at the inflow and outflow of the study area between July 2014 and March 2015 (see van der Deijl et al., 2017). These measurements were used to esti-mate the fraction of sediment (5.8%) the study area has received from the total load of the River Rhine during this period. To calculate the total amount of sediment the study area has received since the opening of the study area, the fraction of 5.8% percent has been multiplied with the total load of the River Rhine, for which data is available for the entire period since the opening of the study area.

We have clarified the method of calculation of the sediment trapping efficiency in both the methods (Section 2.2.4) and the results (Section 3.5): the trapping efficiency was calculated based on the total sediment trapped in the area relative to the total amount of sediment the study area has received since the opening of the study area

**P8 line 7: how is this 46% calculated?**

We have clarified the method of calculation of the sediment trapping efficiency in both the methods (Section 2.2.4) and the results (Section 3.5): the trapping efficiency was calculated based on the total sediment trapped in the area relative to the total amount of sediment the study area has received since the opening of the study area

**P10 line 22: is there seasonality in the sedimentation? What is the seasonality in discharge, in SSC, in tidal amplitude? How much do peak events contribute to**

**the budget? The peak in 2011, was it only a peak in discharge or did this event also had higher SSC?**

In section 2.1 we have added more detailed description of the discharge regime and suspended sediment concentrations of the River Rhine. We also describe that the tide is mixed semidiurnal with a tidal range of 0.2 to 0.4 m. In addition, in section 3.2, we added a sentence that describes the response of the SSC concentration during the 2011 discharge event.

figure-1.png

**Fig. 1.** The study area Kleine Noordwaard, which is located within the Biesbosch Freshwater Tidal Wetland, in the lower Rhine and Meuse delta in the southwest of the Netherlands (a and b). Elevation is shown in meters, with respect to the Dutch Ordnance Datum (NAP) for the period before (c) and after depoldering (d).

figure-2.png

**Fig. 2.** Elevation of Transect A (see Figure 1) with respect to the Dutch Ordnance Datum (m NAP) with subdivision of the area into subtidal areas, flats, low and high marshes, and terrestrial zone relative to mean low water (MLW), mean sea level (MSL), mean high water (MHW), the maximum observed water level (EHW) and the water level for a peak discharge or storm with return period of 1 year, which were used to divide the study area in.

---

## Author Comment (AC2) · 21 Sep 2017

We thank the reviewers for taking the time to read and review our manuscript. We address each specific comment below:

**Reviewer comments are bold**
Author comments are in plain text
The changes in manuscript can be found in the manuscript with track changes

[Figure]

**1 Anonymous Referee #1**

**The authors present results of a comprehensive study on the morphological development of a recently opened wetland, based on an extensive data set.**

**First of all, I completely agree with the questions raised by Reviewer 1.**

For the response to these questions we refer to the document reply to review 1

**The text cites a paper as to be in preparation, by the authors, which is not in the list of references. That, and conference abstracts by the authors, suggest the availability of potentially more data on SSC and current velocities at the entrance to the wetland, which are, in here, deeply missed. I guess, there is a good reason for the suspected splitting of the data set.**

The paper, which was cited as to be in preparation, has meanwhile been accepted and published. Therefore, we have changed all references to this paper in the text to (van der Deijl et al., 2017) and the paper has been added to the list of references.

*van der Deijl, E. C., van der Perk, M., and Middelkoop, H.: Factors controlling sediment trapping in two freshwater tidal wetlands in the Biesbosch area, The Netherlands, Journal of Soils and Sediments, pp. 1–17, doi:10.1007/s11368-017-1729-x, http://link.springer.com/10.1007/s11368-017-1729-x, 2017.*

While this manuscript focuses on the medium-to-long term sediment budget since the opening of the study area , the above paper use 10-minute interval data on water level, flow velocity, and suspended sediment concentrations measured at the inlet and outlet of the study area to identify the controls on sediment trapping. The period for which

these data is available only covers the period July 2014 - March 2015 and represents only a short period since the opening of the study area. This makes this detailed data less suitable for the purpose of this study Nevertheless, we use the data and refer to the above paper in a newly added description of the water levels, discharge, and suspended sediment concentration in the channels of the study area and the inundation frequency of the intertidal flats, marshes and terrestrial zone in the area description (Section 2.1). Furthermore, we used the data for estimating the proportion of the total sediment load of the River Rhine that enters the study area (Section 2.2.4).

**So, since no more information is presented here on the forcing, tide-driven dynamics, etc., the focus should be on the analysis of geospatial information. In this, I think the data source is not (yet) presented to the reader in a proper way. From the suggested high quality of MBES and LIDAR data, I had expected simply better plots, e.g., from one flat in the center of the wetland and surrounding channels, showing more details of aggradation and erosion patters.**

**The color scale in Fig. 1 doesn't help. The interesting elevation range, -2 m to 1 m, is essentially not resolved.**

Figure 1 was included in the manuscript to show both the location of the study area and the transformation of the study area as a result of depoldering, not to derive rates of deposition at the intertidal flats. Because of the regular submergence of the intertidal flats, a complete digital elevation model based on the Lidar data of the intertidal flats is only available before the depoldering of the study area. Therefore, it is not possible to determine rates of deposition at the intertidal flats from different LIDAR datasets.

To further clarify the initial morphology of the study area after depoldering, and the division of the study area in channels, flats, marshes and a terrestrial zone, we have

added a new figure (Fig2).

Furthermore, we added a new figure to the manuscript to further clarify the sedimentation and erosion patters in the channels (Fig 4 in the revised manuscript). This new figure shows the development of channel section 1 and 3 (for location of the transects see Fig 1, which represent the morphological development of both the single channels near the inlet and outlet, and the perpendicular channels in the middle of the study area.

**There are quite some simple methods around, based on gridded spatial data, e.g., the maximum bed elevation range to differentiate between more and less active regions, or, vertical dynamic trend analysis to show - what is already in the text - stagnation of morphological change with time at specific locations. The DEMs could have also been used to extract cross-sections of channels, or longitudinal transects from some channel thalweg, all means to give the reader a good impression of morphological changes. I'm just suggesting. How to proceed, and what to change, depends on the focus.**

In our response to the former comment, we already mentioned that we added a new figure (now Fig 4) to the manuscript to further clarify the aggradation and erosion patters in the channels by showing the morphological development of two channel cross-sections. Furthermore, we changed the data representation in Fig 2 in the first version of the manuscript. The maps in this figure showed the spatial distribution of the total difference in channel bed level. This figure has become Fig 3 in the revised manuscript and the maps now show the difference in channel bed level in m/year (the vertical dynamic trend) for each monitoring period.

**Tja, what is the actual focus?   The determination of the budget?   Analysis of transport both into inside the area? Geomorphological changes of flats, banks, channels? Analysis of the sedimentology, including transport of the coarse and cohesive fractions?**

**I find the focus unclear and the Introduction and Discussion symptomatically unspecific, e.g., page 9 line 24: "cut bank retreat does not significantly contribute to the total sediment budget".   If the paper was about the budget, why is the statement, cut banks would not contribute to that budget, followed by a detailed comparison of cut bank dynamics with literature? That is a question of the focus. The same could be said regarding sand deposition in the deeper channels.**

**The discussion essentially remains on the level of a comparison. In each part of the discussion, process based considerations are used to explain the situation at the specific location.   Which is cool.   But I asked myself at the end of each paragraph, what could be concluded from this discussion for similar geomorphological settings? I had the impression, and maybe I'm wrong, that the discussion did not lead to the conclusions.**

As mentioned in the title of the manuscript, the focus of this paper is "Establishing a sediment budget in a newly opened wetland area".  The rates and patterns of sedimentation and erosion are compared to those in other wetlands to discuss the relative contribution of geomorphological processes in the study area to the budget, and to derive practical implications for the management of newly created tidal freshwater wetlands.

We have made the changes in the introduction to further clarify the focus of the manuscript

In the discussion section, we compare the rates and patterns of sedimentation and erosion to those in other wetlands, and we use the process-based considerations to discuss the relative contribution of geomorphological processes to the sediment budget in our study area. To further clarify the implication of the processes to the sediment budget, each paragraph now ends with a concluding sentence. In sections 4.3 (Implications for management) and 5 (Conclusions) we refer to these conclusions to derive practical implications for the management of newly created tidal freshwater wetlands.

The exact changes in the discussion, can be found in the manuscript with track changes

**The Conclusion is essentially a summary of Results. On one hand, it is said that the wetland is a trap for sediments. On the other hand, it is deduced at least for the channels that the domain is approaching some kind of equilibrium, which certainly didn't exist from the beginning. So, trapping occurs only because of the non-equilibrium state at the opening of the wetland? Which grain sizes could actually be trapped? Is it realistic that sand is transported into the wetland? If trapping was the goal (Paola et al., 2011, is cited in the Introduction), should we conclude that, to efficiently trap sediments, we would need to open new wetlands every once in a while? Ok, seriously, I find this important, as, e.g., around many estuaries we are having this discussion right now, everywhere in Europe.**

We agree that this a topical and relevant discussion. We adapted section 4.3 (Implications for management) by adding reflection on current sedimentation rates and possible measures to enhance these. At the end of the section we state that conversion of polders into wetlands in deltas may be an effective strategy of delta restoration since sedimentation compensates at least partly for sea level rise and land subsidence. These management implications have also been added in the conclusions section.

**So, in summary, I hope for a shorter text, with a clear focus. The plots should reflect the data quality. The Discussion should lead to the Conclusions.**

To recap the response to the former comments: we have adapted and added figures, added concluding sentences in the discussion sections and we refer back to these conclusions in the conclusion.

**Minor comments: P6 L7 An interesting aspect is that "channels cannot migrate freely". Any ideas what this could mean for small-scale morphodynamics and the distribution of sediments? What is that "bank protection"?**

The dikes along the channels near the inlet and outlet of the study area are protected by riprap to prevent erosion. Since the outer bends have reached the riprap, further erosion is prevented. As a result, there is less sediment available for further development of the point bars at the end of the bends and the channels will become fixed.

*However, the channels are not able to migrate freely due to steep banks of dikes armoured by riprap , and the average width to depth ratio of the channel has decreased from 17.9 to 15.2.*

**P6 L23 Any particular reason, why the term "morphodynamic equilibrium" is persistently avoided, using "equilibrium state between their geometry and the flow conditions", instead?**

We had no particular reason to avoid the term morphodynamic equilibrium, which is now used in the manuscript at P7 L26, P10 L1, P10 L9 and P13 L23

**P4 L14 "whether changes . . . had been . . . years"**

We have implemented this comment

**P8 L17 "Yet . . . channels" example of many sentences which can simply be deleted without changing anything in the text**

We have implemented this comment by deleting this sentence.

**P9 L25 "Wind waves . . . fetch" common knowledge, not required**

We have deleted this sentence from the manuscript

**P10 L3 Entire paragraph: example for text with too many assumptions, for my taste, since in the end this does "not allow drawing conclusions"**

We have deleted the last sentence of the paragraph so we now end with the conclusion that resuspension of sediment takes place regularly.

figure-1.png

**Fig. 1.** The study area Kleine Noordwaard, which is located within the Biesbosch Freshwater Tidal Wetland, in the lower Rhine and Meuse delta in the southwest of the Netherlands (a and b). Elevation is shown in meters, with respect to the Dutch Ordnance Datum (NAP) for the period before (c) and after depoldering (d).

figure-2.png

**Fig. 2.** Elevation of Transect A (see Fig. 1) with respect to the Dutch Ordnance Datum (m NAP) with subdivision of the area into subtidal areas, flats, low and high marshes, and terrestrial zone relative to mean low water (MLW), mean sea level (MSL), mean high water (MHW), the maximum observed water level (EHW) and the water level for a peak discharge or storm with return period of 1 year, which were used to divide the study area in.

figure-3.png

**Fig. 3.** The difference in channel bed level and the cumulative channel bed volume for each monitoring period. The cumulative channel bed volume is shown along a N-S transect starting from the Spiering polders (purple in Fig. 1). The budget of the Wassende Maan (blue in Fig. 1) is added at once at the second black line. The channel in the southwest of the study area was dredged in the monitoring period 2012-2015. The dredged area is excluded in the analysis and not shown in the cumulative channel bed volume.

figure-4.png

**Fig. 4.** Bed level of channel section 1 (a) and 3 (b) for all monitoring campaigns. (see Fig. 1 for the locations of the cross sections)

---

## Author Response (AR2)

**Contents of Author's Response**

This document starts with the point-by-point response to the first referee of the second stage of review. This response is followed by a marked-up manuscript , which shows all the changes with respect to the first revision of the discussion paper. These comments and the new version of the manuscript have been taken on board in the evaluation of the second referee in this second stage of review. The second part of this document contains again the point-by-point response to the comments of this referee, and then the marked-up manuscript version showing all the changes with respect to the first second stage revision of the manuscript (mentioned above).

1. response to review 1 round 2
2. manuscript with track changes version 2a
3. response to review 2 round 2
4. manuscript with track changes version 2b

**Response to Reviewer #1, ESurf2017-22 revised**

**This paper has improved a lot. It is much clearer how things are calculated. However, I still have some difficulties with the calculation of the total sediment budget and the trapping efficiency. Although a major focus of this paper is on this trapping efficiency, this is not discussed properly.**

The major focus of this paper is on the sediment budget of the study area. Although we attempted to determine the sediment trapping efficiency by combing the data presented in this manuscript with data from the upstream monitoring station at Lobith, we do not agree that trapping efficiency is a major focus of the manuscript. Given the issues about the calculated trapping efficiency raised by the reviewer (as discussed below), we have decided to remove this calculation of the trapping efficiency from the manuscript and - where relevant – refer to the trapping efficiency derived for the study area reported in a previous study (Van der Deijl et al., 2017)

**The authors now describe the incoming and outgoing discharge at both the north and south inlet of the study site. But what is done with this information? The inflow in the south seems limited: 16 m³/s when water rises rapidly. So this inflow can be ignored? An estimate of this inflow to illustrate this could be useful. In- and outflow discharge is slightly different. Are there other outlets? Is this within the error of the measurements? No info is given**

In response to the reviewers' comments on the previous version of this manuscript, we added the incoming and outgoing discharge and suspended sediment concentrations at the north and south inlet of the study site to the study site description to characterise the area, not to calculate a sediment budget in this paper. Apparently, and unfortunately, reporting the average discharges and sediment concentrations at both the inlet and outlet of the study area gave rise to some confusion because these are slightly different. The reason for this difference is that the water flow reverses at both the inlet and outlet during short periods of time – there are no other outlets in the central part of the study area. The relevance of this in the context of this paper is small, since we did not use these numbers for the calculations of the sediment budget. We have removed the discharges and concentrations for the outlet of the area. Instead, we added the average outgoing discharge in the north and the fraction of the time that outflow at the upstream location and inflow at the downstream location takes place.

**In and outflow discharges and concentrations are given, so based on this, trapping efficiency can easily be made. Why is this not done and discussed? With 89 m³/s and 26mg/l at inflow, and 86m³/s and 19mg/l at outflow, you get 30% trapping, half of what is in this paper. During peak events, with 191 m³/s and 114 mg/l at inflow and 178 m³/s and 62 mg/l, you get 50%. Would be interesting to discuss this.**

> The simple calculation of the sediment trapping efficiency as suggested by the reviewer yields a biased outcome as the reported suspended sediment concentrations are time-averaged concentrations and not discharge weighted concentrations. The trapping efficiency for the period between July 2013 and March 2014 was already derived in the previous study by van der Deijl et al. (2017). We now refer to this trapping efficiency.

**The paper refers to the total load of the River Rhine at the German border. Now some additional information is given, but I still wonder how relevant this is. The SSC increases from 15 mg/l at the border to 26 mg/l at the inlet of the site. So on its way to the site, the water clearly changes a lot. Is there a nice correlation for SSC between both sites? During peak discharges, SSC increases 10fold at the German border, but only 5fold at the study site. How are SSC and sediment load at the border and the site correlated? With the provided information, I have strong doubts whether you can use the average 5.8%, calculated for the period July 2014 – March 2015, to the entire study period to calculate the incoming sediment load. Especially given the major peak discharge in 2011. What was the effect of this for the study site? During this event, the total sediment load passing the border was as big as the load of an entire year!**

> As mentioned above, given the issues and uncertainties of the calculation of the sediment trapping efficiency for the study area based on the measured long-term sedimentation and the data of discharge and suspended sediment at the upstream Lobith monitoring station, we have decided to remove this calculation from the manuscript (both from the methods and results section). We now only refer to the trapping efficiency of the study area as reported by Van der Deijl et al. (2017) (in the introduction and discussion sections).

[revised manuscript text omitted]

**Response to Reviewer #2, ESurf2017-22 revised version 2a - November 2011**

We thank the reviewer for taking the time to read and review our manuscript. We address each specific comment below.

Reviewer comments are **bold**
Author comments are in plain text
The changes in the manuscript are shown in the manuscript with track changes

**P4L28: Can you describe more thoroughly how this correction was calculated? Which data were used to calculate it?**

> We have addressed this comment by adapting the method section. We now give a more detailed description of the method used for the correction of the sediment budget for the dredging in a part of the study area.

**P5L16: This model needs to be explained more. It was not initially clear to me whether you were using a relationship from the literature or one fit to your data.**

> We have further clarified the description of the model for the sedimentation in the intertidal area by adapting the method section. We have added the description of how the fit was determined, and we now explain that the second intercept in the fit represents the Ferguson bias correction factor.

**I do agree with the first anonymous referee that the high heterogeneity of deposition needs to be accounted for in the estimate of the intertidal sediment budget, but I am concerned that the model does not have a lot of predictive power (R2=0.09), so I am not sure how much you gain from using this estimate versus just using the mean of the measurements. Can you quantify the uncertainty in the model prediction? How does that compare to the standard error of the mean of the measured sedimentation rate in the intertidal area?**

> Even though the predictive power (R2) of the model for the intertidal area is only 0.103, the mean squared error of the fit (11.06) is smaller than the mean squared error of the measurements (12.33). Furthermore, the use of the regression model avoids the bias originating from the difference between the distribution of surface elevations of the sample locations and the distribution of the elevations of the entire intertidal area. Therefore, we prefer the use of the model over the use of the mean measured sedimentation.

**P8L19: The value given for the intertidal sedimentation rate here does not match the one given above. It looks like a rounding error.**

> We have adapted the intertidal sedimentation rate in this section. It was indeed a rounding error and the intertidal sedimentation should have been 15.7 $10^3$ m$^3$ year$^{-1}$ (15.74 10^3)

**P8L22: Can you give more detail on how you calculate this future sediment budget? What time scale is this prediction relevant over?**

> We have changed the text. In the new text, we do not make a future projection of the sediment budget, but we limit the description of the results to the calculation of the contemporary sediment budget.

**There is some additional analysis (transport capacity and wind wave/shear stress) in the Discussion. The details of the transport capacity calculations, in particular, could be moved to the Results section, while leaving the key conclusion -- "not only the bifurcation, but also the presence of the wide and shallow intertidal area, results in enhanced sedimentation in the centre of the area" -- in the discussion.**

> We had added the analysis of the transport capacity of the channels to the discussion to indicate the implications for future river delta restoration of the 'designed' dimensions of the channels, and tidal area on both the sediment budget and the distribution of the sediment over the study area.
>
> The inlet of the study area is a very small side channel of the River Nieuwe Merwede, so we had expected a very small transport capacity for this channel relative to the River Nieuwe Merwede, and therefore channel aggradation. However, the inlet channel shows erosion, which occurs when the supply of sediment is less than the transport capacity of the channel. Since the bed level of the inlet is higher than the bed level of the River, we assumed no bed load could enter the inlet of the study area, and that the (2012-2015) sediment budget of the inlet should represent the actual transport capacity, since the inlet channel tends to morphodynamic equilibrium ( in – budget = out, where in = 0).
>
> However, we now understand that these statements are not yet clear in the manuscript and after the reexamination of the development of the morphology of the inlet channel we have decided to remove this analysis, since we also state that the channel bed has eroded to the same depth as the River Nieuwe Merwede, and we see some sedimentation, just after the bifurcation with the River. Therefore, we have decided to remove the second part of the analysis of the total transport capacity from the discussion. It does not affect our overall observations or conclusion.

**The two following statements regarding the transport capacity are unclear to me:**

**P9L10: "the negative sediment budget of the inlet for the period 2012-2015 can be seen as the total maximum transport capacity of this channel." Why is this true? What is the value of the sediment budget/transport capacity?**

> See our response to the previous comment

**P9L13: "the reduced transport capacity of the channels explains only 24% of the positive sediment budget in the centre of the system." I think this means that the reduced capacity of the central channels (54% of whatever the total capacity is) is only 24% of the total sediment budget of the centre of the system. What are these values?**

> See our response to the previous comment

**Figure 5: The years on the x-axis are too closely spaced.**

> We have implemented this comment by increasing the size between the years on the x-axis

**Figure 7: I don't understand why the model equation shows two intercept values. Perhaps this could be accounted for in a more detailed explanation of the model as suggested above.**

> We have further clarified the description of the model for the sedimentation in the intertidal area by adapting the method section. We have added the description of how the fit was determined, and we now explain that the second intercept in the fit represents the Ferguson bias correction factor, which is now also mentioned in the caption of figure 7.

[revised manuscript text omitted]